# Assessing Children’s Dental Age with Panoramic Radiographs

**DOI:** 10.3390/children9121877

**Published:** 2022-11-30

**Authors:** Tal Ratson, Nurit Dagon, Netta Aderet, Eran Dolev, Amir Laviv, Moshe Davidovitch, Sigalit Blumer

**Affiliations:** 1Department of Pediatric Dentistry, Maurice and Gabriela Goldschleger School of Dental Medicine, Sackler Faculty of Medicine, Tel Aviv University, Tel Aviv 69978, Israel; 2Department Oral Rehabilitation, Maurice and Gabriela Goldschleger School of Dental Medicine, Sackler Faculty of Medicine, Tel Aviv University, Tel Aviv 69978, Israel; 3Department Maxillofacial Surgery, Maurice and Gabriela Goldschleger School of Dental Medicine, Sackler Faculty of Medicine, Tel Aviv University, Tel Aviv 69978, Israel; 4Department of Orthodontics, Maurice and Gabriela Goldschleger School of Dental Medicine, Sackler Faculty of Medicine, Tel Aviv University, Tel Aviv 69978, Israel

**Keywords:** dental age assessment, panoramic radiographs, forensic dentistry, the London Atlas

## Abstract

(1) Background: The aim of the study was to assess the dental age of the subjects and compare it to their chronological age; to assess the dominant tooth for evaluation of dental age; and to investigate possible individual differences between the left and right side of the dental arch. (2) Methods: This study involved evaluating panoramic radiographs of patients aged 7–13 years. A separate assessment was performed for each tooth according to the degree of germ development. Each subject’s dental age was estimated. (3) Results: The study involved evaluating 349 panoramic radiographs. No difference was found between stages of tooth development on the right side and left side. Correlation between the stages of tooth development and the chronological age was found to be highest in the second permanent molar teeth. The age can be predicted with the estimated age of tooth #37 and the gender of the patient. (4) Conclusions: The chronological age of children aged 7–13 may be estimated based on a modified seven-stage London Atlas of tooth development, where the most accurate landmark(s) of use are second molars.

## 1. Introduction

One of the most important characteristics used to establish the identity of any individual in different legal, forensic, or anthropological research contexts is age estimation. Moreover, age estimation is increasingly requested by the judicial system for establishing immutability, and in the case of Juvenile Court, determining the age of undocumented adopted minors [1,2].

There are many indicators for age assessment. Among them are skeletal maturity, body height and weight, sexual development, and tooth development and eruption [3].

Dental age assessment can be performed by various methods: visual, radiological, morphological, biochemical, and histological.

The visual method evaluates: 1. The sequence of tooth eruption: this may provide dental age assessment up to 12–13 years of age. This method is considered not reliable since it is influenced by local and systemic factors. 2. Tooth structure: wear and attrition, tooth color and stains. This method is also considered less reliable. 

Radiographic methods are based on the evaluation of tooth development on the various radiographic images to assess the degree of tooth mineralization from the moment when radiopaque spots become visible prior to tooth calcification until the tooth apex is closed. This method allows continuous assessment of tooth development from birth until the completion of third molar teeth development. This method assesses the stages of tooth formation from the initial mineralization of a tooth, the crown formation, root growth, eruption of the tooth into the mouth, and root apex maturation. It is mainly suitable for children–adolescents. It is a simple, noninvasive, and reproducible method that can be employed both on living and unknown dead individuals [4].

There are many radiographic images that can be employed in dental age assessment, including periapical radiographs, lateral oblique radiographs, cephalometric radiographs, panoramic radiographs, digital imaging, and advanced imaging technologies. The radiographic images must include developing relevant teeth, and all the stages of dental development can be rated [5]. Today, the most available, immediate, and accessible tool on offer is the panoramic radiograph. This imaging requires less X-ray radiation compared to status imaging, is easy to perform, and does not involve any discomfort to the patient. 

Dental age assessment is one of the most rapid and reliable scientific methods, since dentition is resilient to nutritional, hormonal, and pathological changes, particularly in children [5]. Dental age assessment is usually required for the identification of victims of mass disasters, or living individuals; mainly in proceedings concerning adoptions, migrations, juvenile abuses, legal consent, and some other medico-legal cases [6]. Dental age assessment has been adopted within pediatric dentistry and orthodontics as diagnostic aids, particularly in the context of expected maxillofacial growth. 

Dental age assessment in children and adolescents is mainly performed either by using the atlas approach or by using scoring systems. In the atlas method, morphologically distinct stages of mineralization shared by all teeth are observed and compared to a representative compendium (atlas), corresponding to chronological age; in the scoring method, dental development is divided into different stages that are assigned maturity scores for each tooth, evaluated through statistical analysis, and then compared to known age standards [4].

The purposes of the present study were to assess the dental age of each subject and compare it to their chronological age, in order to evaluate the dominant tooth for evaluation of dental age and to investigate possible individual differences between the left and right side of the dental arch. 

## 2. Materials and Methods

This cross-sectional study involved the evaluation of 363 consecutive panoramic radiographs from patients aged 7–13 years, based on the university database kept from 1997 to 2021. All the individuals involved gave their informed consent. The study was approved by the ethical (Helsinki) committee of the university where the study was carried out, and it was conducted in full accordance with the World Medical Association Declaration of Helsinki (Approval No. 120.19). 

Inclusion criteria were the following: subjects’ age range being from 7 to 13 years-old; high-quality radiographs with detailed recording of date of birth and date that the radiograph was performed; healthy subjects without systemic or syndromic pathologies; and normal dentition without any dental abnormalities. Exclusion criteria were the following: radiographs that were unclear; subjects with systemic diseases (e.g., hyperthyroidism, hypothyroidism), syndromes (e.g., Down’s syndrome, ectodermal dysplasia), congenital developmental abnormalities (e.g., agenesis or supernumerary); and impacted or ankylosed teeth. Fourteen radiographs were deemed unqualified and rejected for inclusion in this study. Therefore, the final study sample was comprised of 349 such panoramic radiographs.

Dental age assessment was performed by a single examiner (N.A.), a resident in the pediatric dentistry department under the supervision of senior staff members from the pediatric dentistry and orthodontic departments. The chronological age of the subjects was determined by subtracting their birth date from the date on which radiographs were taken. A separate assessment was performed for each tooth according to the degree of crown and root development. The main examiner received the radiographs without any other information and performed the dental age assessment blind to chronological age.

The assessment was performed according to the tables of the London Atlas [7]. To estimate age with the London Atlas method, radiographs were assessed to identify the developmental and growth stages for permanent teeth, excluding central and lateral incisors and the first permanent molar due to their early apex closer (usually by the age of 9 years). 

The original table presented as part of the London Atlas includes 13 delineated stages of tooth development, but it was difficult to demarcate the different stages on a panoramic radiograph. Therefore, the 13 stages were consolidated into seven more distinguishable stages (similar stages were omitted) (Table 1) [7].

Each subject’s dental age was assessed and compared to their chronological age, and comparative statistics were performed. Furthermore, differences between each tooth on the left and right side of the dental arch were examined.

### Statistical Analysis

All statistical analyses were performed using SPSS software (IBM SPSS Statistics for Windows, version 24.0, IBM Corp, Armonk, NY, USA). A *t*-test was used to compare males and females (with normal distribution). Pearson correlation was used to analyze the correlation between the right and left side, as well as the correlation between chronologic age and tooth developmental age. A *p*-value of <0.05 was considered statistically significant.

## 3. Results

The study involved evaluating 349 qualified repetitive panoramic radiographs from a university archive. There were 191 females with a mean age of 12.30 ± 2.32 and 158 males with a mean age of 12.40 ± 2.16. There was no statistical difference in chronological age between males and females in these groups (*p* = 0.678).

Table 2 shows the age and the different teeth analyzed, comparing females and males. It was found that there are statistically significant differences between males and females in teeth #13, 14 and 15 with no clinical significance (11.23 compared to 10.99, 10.39 compared to 9.91, and 10.69 compared to 10.52, respectively). When the right and left homologous teeth in the upper and lower jaw were separately compared, there was a statistically significant correlation found between all paired teeth (r > 0.9, *p* < 0.0001). That is, there was no difference found between stages of tooth development on the right and left side.

It was discovered that the correlation between the stages of tooth development and the chronological age of each subject was statistically significant for all teeth (Table 2). This correlation was found to be highest in the second permanent molar teeth (r = 0.764, 0.766, 0.749, and 0.748 for teeth # 17, 27, 37, and 47, respectively).

A stepwise linear regression model was used in order to predict the chronological age of the given subjects, based on the analyzed data (Table 3). A significant regression equation was determined based on tooth development and gender with R^2^ = 0.64. The age can be predicted with the estimated age of tooth #37 and the gender of the patient.

## 4. Discussion

Dental age assessment methods are of great value, since teeth are remarkably resistant to mechanical, chemical, or physical impacts and time. Furthermore, dental age is minimally influenced by nutritional, medical, environmental, and living conditions. The various methods used for dental age assessment in individuals include radiological methods, as opposed to histological and biochemical methods. The latter methods involve either extraction or preparation of microscopic sections of at least one tooth from a given subject in order to acquire chronological age. The radiographic method, on the other hand, is a simple, quick, economic, and minimally invasive method of age identification, which can be used as a diagnostic tool for estimating the age of either deceased or living subjects across all communities [8].

Moorrees et al. first published their standardized use of such a method in 1963 [9], which detailed 14 stages of dental development of the permanent dentition as determined from radiographic evidence, and the cumulative assessment derived from this method determined the mean age for the corresponding stage. The youngest age in their survey was 6 months and the records included the development of the third mandibular molars. Notably, female development was more advanced of that of males and the root formation stages showed higher variability compared with stages of crown formation [9]. 

This study was based on the London Atlas (Queen Mary Innovation Ltd., London, UK) [7]. AlQahtani et al. published this novel dental age assessment method in 2010. The London Atlas is a pictorial textbook that involves the assessment of stage formation and eruption for each tooth and then matches it to one of the 31 illustrations of age categories representative of both tooth formation and tooth eruption. The tooth formation stages were amended from Moorrees et al. [9]. The London Atlas was initially tested on subjects of British and Bangladeshi ethnicity and proved to be more precise than Schour and Massler’s and Ubelaker’s methods and potentially more accurate than Demirjian’s calculations [10,11,12,13].

In the present study, there was no statistically significant difference found between estimated age and actual chronological age, as determined from either the left or the right sides of the jaw. This finding is similar to that previously reported in Portuguese and Brazilian children [14,15].

All teeth that were examined showed a statistically significant correlation to the chronological age of each subject. Furthermore, the present study determined that the most accurate correlation between chronological and dental age in children aged 7–13 was found when relying on the second permanent molar teeth (17, 27, 37, and 47). This determination has not been described elsewhere in the literature and seems to be a unique outcome of this study. Although a high correlation between the upper canines’ tooth age and chronological age was found, the fact that the number of teeth assessed was considerably lower that the number of the second molars aimed the focus on the second permanent molars. This lower number of upper canines assessed was due to the fact that, in many cases, the upper canines’ roots are obscured in the panoramic X-ray.

Comparing the findings by gender indicates that it is necessary to subtract six months from the age estimation of females to better approximate their actual biological/chronological age. This finding is similar to that previously reported in other studies, which indicated that female development was ahead of that of males [14,16]. In a meta-analysis by Franco et al. published in 2020, it was also reported that female age could be estimated more accurately than male age using seven different methods of age estimation [17].

Dental age assessment plays a major role in many fields. The conclusions arising from this research give the clinician or researcher the ability to assess dental age while relying on a seven-stage scale of tooth development of a specific tooth (i.e., the second permanent molar). The clinical applications of this research may be utilized in expediting dental age assessment in clinical settings in the pediatric dentistry or orthodontic fields. In the legal or forensic fields, it can assist in assessing dental age using only a single tooth.

The present study has some limitations. Specifically, since the data were derived from subjects treated at a university clinic, there may have been some biases in certain clinical findings. In addition, using panoramic radiographs created difficulty in observing the landmarks due to root overlap/crowding, so it was difficult to assess the root development in some cases; these cases occurred more often in the maxillary arch, in particular, at the canine and premolars region. Our need to consolidate the original 13 stages of tooth development used in the London Atlas to seven may have blurred some of their distinctions. Notwithstanding those issues, the results of our study, using a modified seven-stage London Atlas of tooth development, were similar to those of other studies [14,15,18].

## 5. Conclusions

The chronological age of Israeli children aged 7–13 may be estimated based on a modified seven-stage London Atlas of tooth development, where the most accurate landmarks are second molars. Within this age range, female dental age is 6 months ahead of male dental age when the second permanent molar is used to determine age. This can serve clinicians and investigators in assessing dental age relying on a single tooth.

The differences in age assessment between males and females indicates that universal charts and schemes cannot be used accurately to evaluate both genders. Therefore, this study suggests that separate methods of comparative measurement be developed for each gender.

## Figures and Tables

**Table 1 children-09-01877-t001:** A modified seven-stage London Atlas of tooth development for single-rooted and multirooted teeth [7].

Stage	Ri	R ¼	R ½	R ¾	Rc	A ½	Ac
Single root tooth	Initial root formation with divergent edges	Root length less than crown length	Root length equals crown length	Three quarters of root length with diverge ends	Root length completed with parallel ends	Apex closed (root ends convergence) with wide PDL	Apex closed with normal PDL width
Multirooted tooth	Initial root formation with divergent edges	Root length less than crown length with visible bifurcation area	Root length equals crown length	Three quarters of root length with diverge ends	Root length completed with parallel ends	Apex closed (root ends convergence) with wide PDL	Apex closed with normal PDL width

**Table 2 children-09-01877-t002:** Statistics for examined teeth.

Tooth No	Gender	Number of Teeth	Mean Tooth Age + SD	*p* Value for Gender Difference	Correlation between Tooth Age and Chronological Age	*p* Value for Correlation to Age
13	F	111	11.23 ± 0.77	0.046	0.752	<0.0001
	M	83	10.99 ± 0.87			
23	F	113	11.25 ± 0.77	0.060	0.771	<0.0001
	M	87	11.03 ± 0.86			
33	F	67	9.34 ± 0.87	0.244	0.594	<0.0001
	M	71	9.16 ± 0.94			
43	F	67	9.32 ± 0.89	0.186	0.587	<0.0001
	M	69	9.11 ± 0.97			
14	F	66	10.39 ± 1.03	0.041	0.716	<0.0001
	M	39	9.91 ± 1.31			
24	F	65	10.43 ± 1.06	0.238	0.735	<0.0001
	M	41	10.16 ± 1.28			
34	F	84	10.56 ± 1.05	0.486	0.670	<0.0001
	M	72	10.42 ± 1.27			
44	F	86	10.15 ± 0.98	0.168	0.629	<0.0001
	M	72	9.92 ± 1.14			
15	F	85	10.59 ± 0.81	0.174	0.697	<0.0001
	M	56	10.39 ± 0.87			
25	F	80	10.51 ± 0.87	0.112	0.708	<0.0001
	M	57	10.26 ± 0.94			
35	F	120	10.68 ± 0.69	0.147	0.706	<0.0001
	M	92	10.52 ± 0.84			
45	F	123	10.69 ± 0.71	0.025	0.711	<0.0001
	M	96	10.45 ± 0.88			
17	F	135	10.54 ± 0.99	0.255	0.764	<0.0001
	M	101	10.38 ± 1.14			
27	F	135	10.55 ± 1.01	0.237	0.766	<0.0001
	M	101	10.38 ± 1.14			
37	F	136	10.60 ± 0.97	0.149	0.749	<0.0001
	M	102	10.41 ± 1.11			
47	F	134	10.58 ± 0.98	0.213	0.748	<0.0001
	M	101	10.41 ± 1.12			

F-Female, M-Male.

**Table 3 children-09-01877-t003:** Stepwise linear regression model.

	β	*p*-Value
Constant	4.168	0.004
Tooth #37	0.603	<0.0001
Gender-Female	−0.538	<0.0001

## Data Availability

Data are available on request due to ethical restrictions.

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
