# Peer review of "Assessing Children’s Dental Age with Panoramic Radiographs"

_children, 2022, doi:10.3390/children9121877_

Round 1

Reviewer 1 Report

Dear authors,

please verify the total number of panoramic images examined: if 348 or 349. Please specify in 'material and methods' the number of observers and how you managed to bypass any bias in the final assessments. 

In the 'discussion' I suggest to highlight possible clinical applications of your research.

Author Response

Dear reviewer
We would like to thank you for your time and effort that you took in reviewing our manuscript.
We have read your remarks, and edited our manuscript, Please see the attachment.
We hope you will find it satisfactory.
Please let us know if there is anything else that needs our attention.
Sincerely yours
The Authors

Reviewer 2 Report

The authors investigated the relationship between dental age and chronological age in panoramic radiography. The study design and results showed almost clearly but needs some revision. My comments are summarized as follows. 

Comments

Material and Methods

1.       Please note that the observer’s number and status (what kind of field in expert, how many years).

2.       The most matter of concern is accuracy of dental age staging.

-          How were staging repeatability and concordance? This is the most important issue for the reliability of your results. It is difficult for me to divide into R 3/4, Rc, A 1/2, Ac accurately only using panoramic radiography (in Figure 1).

-          Please note the details of panoramic features to estimate dental age.

Results

3.       It seems to have highest correlation not only second molar but also maxillary canines. Why the canines excluded to be used for estimation?

Figure

4.       The resolution of the images and explanations are too low. Please show clearer one.

5.       It should be displayed with panoramic images for readers’ easy understanding.

6.       I suggest adding another figure to show the deference between male and female dental age (summarized same chronological male and female panoramic images).

Author Response

(The authors gave the same response as above.)

Reviewer 3 Report

Dear authors,

The abstract should provide more concrete data. Avoid general sentences…i.e

”Differences between the left and right side of the dental arch were examined”

“Correlation between the stages of tooth development and the chronological age was statistically significant for all teeth”

Remove lines 31-37

The ethical approval number of the study should be provided in the material and method section

The inclusion criteria should be better explained: please refer to the age of the participants and to the panoramic radiography pathologies as well

Also, exclusion criteria cannot be only poor quality radiographs, systemic diseases or dental abnormalities. Please expand these criteria to be more specific.

“Thirteen radiographs were deemed unqualified and rejected for inclusion in this study” – is unclear – please reconsider

The assessment of dental age according to Moorrees should be explained

If figure 1 is after [8], written permission for reproduction should be provided

Statistical analysis should be better described, and normal distribution should be emphasized. Also, sample size calculation should be provided. What is the NNT?

“It must be stated that a small number of data points could not be ascertained because they were obscured by severely overlapping teeth.” is unclear – please rephrase

Table 1 should be further described in the results section

What is the clinical relevance of “It was discovered that the correlation between the stages of tooth development and the chronological age of each subject was statistically significant for all teeth”

A stepwise linear regression model should be explained in the results section. Why did you perform it?

A significant regression equation was determined based on tooth development and gender with R2=0.64 – is also unclear. What is the relevance?

The material and method section must be better described. It is unclear how and where you performed investigations.

Table 1 and 2 captions should be rewritten. The “The constant as well as the included 148 parameters (tooth #37 and gender) were statistically significant” does not necessary need to be emphasized in the caption

The discussion should focus on the relevant similitudes and differences between the manuscript and the literature.

Limitations of the study should be provided

The conclusion should be rewritten to highlight the relevant findings – and show the clinical relevance. It has to be sustained by the results.

There are many references before 2000. Please reconsider

Author Response

(The authors gave the same response as above.)

Round 2

Reviewer 3 Report

Congratulations on your work!